# MicroRNA-29a Mitigates Subacromial Bursa Fibrosis in Rotator Cuff Lesion with Shoulder Stiffness

**DOI:** 10.3390/ijms20225742

**Published:** 2019-11-15

**Authors:** Jih-Yang Ko, Wei-Shiung Lian, Tsai-Chen Tsai, Yu-Shan Chen, Chin-Kuei Hsieh, Chung-Wen Kuo, Feng-Sheng Wang

**Affiliations:** 1Departments of Orthopedic Surgery, Kaohsiung Chang Gung Memorial Hospital, Kaohsiung 83301, Taiwan; kojy@cgmh.org.tw (J.-Y.K.); m94h0219@gmail.com (T.-C.T.); 2Center for Shockwave Medicine and Tissue Engineering, Kaohsiung Chang Gung Memorial Hospital, Kaohsiung 83301, Taiwan; 3Department of Medical Research, Kaohsiung Chang Gung Memorial Hospital, Kaohsiung 83301, Taiwan; lianws@gmail.com (W.-S.L.); ggyy58720240@gmail.com (Y.-S.C.); jorno0329@gmail.com (C.-K.H.); 4Core Lab for Phenomics& Diagnostics, Kaohsiung Chang Gung Memorial Hospital, Kaohsiung 83301, Taiwan; 5Graduate Institute of Clinical Medical Sciences, Chang Gung University College of Medicine, Kaohsiung 83301, Taiwan

**Keywords:** miR-29a, fibrosis, subacromial bursa, shoulder stiffness

## Abstract

Rotator cuff lesion with shoulder stiffness is a major cause of shoulder pain and motionlessness. Subacromial bursa fibrosis is a prominent pathological feature of the shoulder disorder. MicroRNA-29a (miR-29a) regulates fibrosis in various tissues; however, the miR-29a action to subacromial bursa fibrosis remains elusive. Here, we reveal that subacromial synovium in patients with rotator cuff tear with shoulder stiffness showed severe fibrosis, hypertrophy, and hyperangiogenesis histopathology along with significant increases in fibrotic matrices collagen (COL) 1A1, 3A1, and 4A1 and inflammatory cytokines, whereas miR-29a expression was downregulated. Supraspinatus and infraspinatus tenotomy-injured shoulders in transgenic mice overexpressing miR-29a showed mild swelling, vascularization, fibrosis, and regular gait profiles as compared to severe rotator cuff damage in wild-type mice. Treatment with miR-29a precursor compromised COL3A1 production and hypervascularization in injured shoulders. In vitro, gain of miR-29a function attenuated COL3A1 expression through binding to the 3’-untranslated region (3′-UTR) of COL3A1 in inflamed tenocytes, whereas silencing miR-29a increased the matrix expression. Taken together, miR-29a loss is correlated with subacromial bursa inflammation and fibrosis in rotator cuff tear with shoulder stiffness. miR-29a repressed subacromial bursa fibrosis through directly targeting COL3A1 mRNA, improving rotator cuff integrity and shoulder function. Collective analysis offers a new insight into the molecular mechanism underlying rotator cuff tear with shoulder stiffness. This study also highlights the remedial potential of miR-29a precursor for alleviating the shoulder disorder.

## 1. Introduction

Rotator cuff lesions with shoulder stiffness is a common cause of pain, limited motion, and poor biomechanical function of shoulder, terribly impacting patients’ daily life and activity [1,2]. While the etiological factors remain uncertain [3], subacromial bursitis, like hypercellularity [3], degeneration [4], and hypervascularity [5], is a prominent feature of the shoulder disorder. We have previously revealed that chronic inflammation is correlated with rotator cuff lesions with shoulder stiffness in patients with diabetes [6]. Myofibroblasts overgrowth within subacromial bursa compartment is a notable pathohistological feature in the injured shoulders [7]. Still, little is known about how fibrosis takes place in this shoulder disease. 

MicroRNAs, small non-coding RNA with around 22 nucleotides, are shown to regulate mRNA expression post-transcriptionally during tissue development and damage [8,9]. With regard to microRNA actions to fibrotic responses, increasing reports show that many microRNAs, like miR-1, miR-133, miR-206 and miR-124, are involved in inflammatory myopathy [10] and fibrotic matrix deposition in pulmonary vascular fibroblasts [11]. miR-145, miR-223 and miR-494 regulate cystic fibrosis of airway epithelium [12]. miR-124 knockdown alters TGF-β1 signaling pathways, attenuating extracellular matrix production in renal tissue environments [13]. 

Of microRNAs, the microRNA-29 (miR-29) family is shown to inhibit fibrogenic factors or extracellular matrix production [14]. Loss of miR-29 function increases fibrosis-promoting factor production, like matrix metalloproteinases, ADAM, and fibronectin, in renal tissue microenvironment [15] and promotes the shift of hepatic stellate cells into myofibroblasts in the development of liver fibrosis [16]. On the contrary, increasing miR-29 improves bleomycin-induced pulmonary tissue fibrosis [17] and carbon tetrachloride-mediated hepatic fibrosis [18]. We have previously uncovered that miR-29a was indispensable in preventing synovial fibrosis in the development of human osteoarthritis [19]. miR-29a transgenic mice showed minor response to bile duct ligation-induced liver fibrosis [20] and hyperglycemia-mediated kidney fibrosis [21]. miR-29 also controls collagen synthesis in human tendinopathy [22]. The biological function of miR-29a to subacromial bursa fibrosis in rotator cuff lesion with shoulder stiffness warrants characterization. 

This study aimed to investigate whether the miR-29 family is relevant to the development of subacromial bursa fibrosis in rotator cuff lesion with shoulder stiffness and characterize how miR-29a regulated fibrotic matrix accumulation in inflamed human tenocytes.

## 2. Results

### 2.1. Subacromial Bursitis in Rotator Cuff Lesion with Shoulder Stiffness

First, we examined whether inflammation or fibrosis in subacromial bursa was correlated with rotator cuff tear with shoulder stiffness. We enrolled patients with rotator cuff lesion with (*n* = 22) and without (*n* = 35) shoulder stiffness who had symptoms of impingement for over 3 months along with magnetic resonance imaging (MRI) of complete rotator cuff tear, as well as required surgery of open acromioplasty, lysis of adhesions, or rotator cuff repair [6,7]. In the stiffness group, patients’ gender, involved side, age, and BMI were comparable to the non-stiffness group, whereas shoulder function was significantly downregulated as evident from significant reductions in functional score of Constant and Murley and shoulder range of motion (Table 1). Subacromial synovial specimens from greater tuberosity to coracoid within rotator cuff lesion in both groups were harvested for histopathology assays during surgery.

Subacromial bursa in the stiffness group showed synovial membrane hypertrophy as evident from hematoxylin and eosin staining. Fibrotic tissue exhibited blue and non-fibrotic tissue displayed red in Masson’s trichrome-stained specimens. Specimens in the stiffness group showed spacious fibrotic tissue formation as compared to the non-stiffness group. Vessel overdevelopment also occurred in the stiffness group as evident from strong α-smooth muscle actin (α-SMA) immunostaining (Figure 1A). Consistently, synovial membrane thickness (Figure 1B), fibrotic tissue area (Figure 1C), and vessel number (Figure 1D) were significantly increased in the stiffness group. In addition, inflammatory cytokines, like interleukin (IL)-1β (Figure 1E), IL-6 (Figure 1F), and IL-8 (Figure 1G) expression along with fibrotic matrices collagen (COL)1A1 (Figure 1H), COL3A1 (Figure 1I), COL4A1 (Figure 1J), and ADAM12 (Figure 1K) expression were significantly increased in the stiffness group.

### 2.2. Decreased miR-29a Expression in Rotator Cuff Lesion with Shoulder Stiffness

Given that the miR-29 family members, like miR-29a, miR-29b, and miR-29c, are shown to regulate tissue fibrosis [15,16,17,19], we investigated whether the miR-29a family was relevant to rotor cuff tear with shoulder stiffness. miR-29a and miR-29b rather than miR-29c expression were significantly downregulated in subacromial bursa in the stiffness group (Figure 2A). Serum miR-29a levels were also reduced in the stiffness group, whereas serum miR-29b or miR-29c levels were comparable to the non-stiffness group (Figure 2B). Consistently, cells in subacromial bursa in the stiffness group showed weak miR-29a transcripts as evident from in situ hybridization (Figure 2C).

### 2.3. miR-29a Overexpression Attenuated Inflammation and Gait Irregularity

The analysis of miR-29a loss in subacromial bursa fibrosis in human shoulder stiffness prompted us to utilize mice overexpressing miR-29a (miR-29aTg) (Figure 3A) to test whether miR-29a changes fibrosis during rotator cuff lesion. We adopted supraspinatus and infraspinatus tenotomy-induced rotator cuff injury in mice to mimic excessive inflammation or irregular gait profile of injured shoulders [23] (Figure 3B). In wild-type mice, injured shoulders showed increased echogenicity at 12 weeks postoperatively as evident from sonography, which was suggestive of tissue fibrosis or swelling. The ultrasound echo signal was reduced in miR-29aTg mice (Figure 3C). We adopted fluorescence 2-deoxyglucose injection together with in vivo near-infrared fluorescence imaging approaches, which have been employed to trace inflammation in atherosclerotic vessels, osteoarthritic synovium, and acute pancreatitis [24,25,26], and investigated whether miR-29a overexpression changed inflammation of injured rotator cuff. Injured shoulders in wild-type mice exhibited significant increases in fluorescence reactions, whereas miR-29aTg mice showed mild fluorescence signal (Figure 3D). In addition, shoulder injury resulted in irregular footprint of forelegs in wild-type mice (Figure 3E) along with significant reductions in standing time, maximum contact area, footprint length, footprint area, swing speed, and duty cycle (Figure 3F). miR-29a overexpression improved the footprint histograms and gait profiles of the forelegs with injured shoulders.

### 2.4. miR-29a Overexpression Downregulated Fibrotic Matrix Formation

Of interest, the shoulder lesion in wild-type mice showed strong Masson’s trichrome staining (Figure 4A). Tissue around injured tendons in wild-type mice also displayed strong vessel marker α-SMA immunostaining (Figure 4B) and fibrotic matrix COL3A1 immunoreactivity (Figure 4C). In miR-29aTg mice, injured tendons exhibited mild response to fibrotic matrix and capillary vessel formation together with significant reductions in fibrotic tissue area, vessel number, and COL3A1 expression in the injured tendon (Figure 4D).

### 2.5. miR-29 Precursor Treatment Improved Tendon Fibrosis in Injured Shoulders

Given that rotator cuff injury was improved in miR-29aTg mice, we further tested whether administration with miR-29a precursor changed the tenotomy-induced fibrosis or shoulder dysfunction. Lentivirus human miR-29a precursor or lentivirus empty vector as mock was injected into the injured site in wild-type mice under sonography guide at 1 week postoperatively (Figure 5A). Treatment with lentivirus miR-29a precursor significantly increased serum miR-29a levels, whereas mock injection did not significantly change serum miR-29a level as compared to control group (Figure 5B). Of interest, the treatment compromised echogenicity of lesion (Figure 5C) along with improved footprint histograms of the forelimbs with injured shoulders (Figure 5D). Standing time, footprint length, maximum contact area, and footprint area of forelegs with injured shoulders were significantly improved in the miR-29a precursor-treated group (Figure 5D). Likewise, injured tendons in the miR-29a-treated group showed weak α-SMA (Figure 6A) and COL3A1 immunostaining (Figure 6B) along with minor fibrotic tissue formation as evident from Masson’s trichrome staining (Figure 6C). miR-29a precursor treatment significantly attenuated the tenotomy-augmented vessel formation and fibrotic tissue development (Figure 6D). 

### 2.6. miR-29a Directly Targeted COL3A1 Transcription in Inflamed Tenocytes

We deciphered how miR-29a expression was reduced and how it affected fibrotic matrix formation in tendon tissue. To this end, human tenocytes were incubated in inflammatory cytokines to mimic inflamed fibroblasts in subacromial bursitis. IL-1β rather than IL-6 and IL-8 treatment significantly reduced miR-29a expression in tenocytes (Figure 7A). miR-29a precursor transfection increased miR-29a expression (Figure 7B) but significantly attenuated COL3A1 expression (Figure 7C) in inflamed cells. On the contrary, miR-29a antisense oligonucleotide (miR-29a-AS) transfection decreased miR-29a expression and significantly increased COL3A1 expression in tenocytes without IL-1β stress. Bioinformatics (www.targetscan.org) showed that COL3A1 is a putative target of miR-29a. We constructed luciferase reporters for wild-type and 3-base pair mutated 3′-UTR of COL3A1 (Figure 7D). Of interest, increasing miR-29a significantly decreased luciferase reporter activity of 3′-UTR of COL3A1, whereas miR-29a-AS upregulated the activity in tenocytes. miR-29a did not significantly change the mutated 3′-UTR reporter reaction of COL3A1, which suggests that miR-29a binds to the 3′-UTR of COL3A1 (Figure 7E). miR-29a precursor also attenuated fibrotic marker expression, like smooth muscle actin (SMA), disintegrin and metalloproteinase domain 12 (ADAM12), and 2-oxoglutarate 5-dioxygenase 2 (PLOD2), in inflamed tenocytes (Figure 7F). miR-29a-AS significantly augmented SMA, ADAM12, and PLOD2 expression in tenocytes without IL-1β stress (Figure 7F). 

## 3. Discussion

Rotator cuff lesion with shoulder stiffness provokes excoriating pain and limited motion of shoulders, hugely devastating patients’ life quality. Trauma, diabetes, dyslipidemia, and smoking, etc. put shoulders at the risk of the disease [27,28,29]. Rehabilitation [30], arthroscopic repair [31], and intra-articular injections of steroid [32] are common modalities. Hyaluronic acid [33] and platelet-rich plasma injections [34] improve rotator cuff lesions in experimental animals. Increasing evidence reveals that subacromial bursitis is relevant to shoulder stiffness; however, the molecular mechanism underlying the pathological feature has not been fully understood. This study is the first indication that miR-29a signaling loss was correlated with subacromial bursitis in the development of rotator cuff tear with shoulder stiffness. miR-29a was indispensable in repressing fibrotic matrix COL3A1 overproduction in inflamed subacromial bursa. This study sheds new light to the microRNA pathway attenuation of fibrosis in subacromial bursa compartment during rotator cuff lesion with shoulder stiffness. We also convey a microRNA remedial potential for slowing subacromial bursa deterioration to alleviate shoulder dysfunction. 

Patients with rotator cuff tears with shoulder stiffness had limited motion of shoulder joints, like significant reductions in functional score of Constant and Murley together with SROM. Our analysis was in agreement with other groups’ investigations showing that patients inflicted with this disorder, independent of the gender, BMI, and involved side, underperformed in shoulder function [35,36,37]. While the cause of shoulder stiffness remains unclear, increasing evidence reveals that defective subacromial bursa integrity is associated with the shoulder disorder. This feature prompted us to understand what deleterious activity in the subacromial bursa microenvironments occurred in the development of shoulder stiffness. 

Severe subacromial bursitis signs, like swelling and capillary vessel formation along with fibrotic tissue, overdeveloped in the stiffness group. Significant increases in inflammatory cytokine and fibrotic matrix collagen expression also consolidated the histopathological investigations. Sonography [38] and magnetic resonance imaging [39] show that subacromial bursa thickening is correlated with impingement and rotator cuff tear; however, the cellular events underlying rotator cuff tear with shoulder stiffness have not yet been fully investigated. This study conveyed emerging evidence of fibrotic matrix production in subacromial bursa microenvironment, which exacerbated shoulder stiffness. Fibroblasts in tendon and muscle tissues are shown to overproduce extracellular matrices when they are dysregulated by fibrosis-regulatory factors [40]. Inflammatory cytokines, transforming growth factor-beta 1, and bone morphogenetic proteins are shown to modulate these deleterious reactions [41]. The candidate molecule protective against subacromial bursa fibrosis warrants investigation.

Increasing evidence reveals that microRNA signaling is relevant to tenocyte behavior [42] and ACL-induced tendinopathy [43]. Of microRNA, the miR-29 family is shown to regulate fibrotic matrix expression in IL-33-stressed human tenocytes [22] and collagen production in various tissues in pathological states [19,20,21,22]. In this study, downregulated miR-29a expression in subacromial bursa and serum was correlated with fibrosis in human rotator cuff lesion with shoulder stiffness. In experimental rotator cuff injury models, miR-29aTg mice showed moderate response to tenotomy-mediated hyperangiogenesis, fibrosis, and COL3A1 synthesis along with irregular walking patterns of forelimbs with injured shoulders. miR-29a silencing provoked COL3A1 expression in human tenocytes, whereas miR-29a precursor transfection compromised fibrosis in inflamed tenocytes. The investigations of cell cultures and experimental animals were consistent with the analysis of clinical specimens. Experimental analyses of this study were in agreement with other studies reporting that miR-29a reduces fibrotic responses in pulmonary sclerosis, [17], toxin-mediated liver injury [18], osteoarthritic synovitis [19], and diabetic kidneys [21], etc. COL3A1 is an important hallmark of fibrosis in chronic rotator cuff tears [44,45]. In this study, miR-29a directly targeted COL3A1 3′-UTR together with decreased fibrosis marker expression, which underpinned the analysis of fibrosis-inhibitory action of miR-29a in subacromial bursa. We do not exclude the possibility that miR-29a may target other fibrogenic factors or fibrotic matrices in rotator cuff injury. The anti-fibrosis spectrum of miR-29a in rotator cuff injury warrants further investigations. This study offered a microRNA insight into how miR-29a repressed subacromial bursa fibrosis during rotator cuff lesions of shoulder stiffness. Of note, treatment with miR-29a precursor downregulated the tenotomy-mediated fibrosis, subsequently improving gait characteristics of animals with injured shoulders. miR-29a administration is shown to slow superficial digital flexor tendon injury in horse [46]. This study highlights the remedial effect of miR-29a precursor treatment for delaying fibrosis in shoulder stiffness. 

We acknowledge the limitation of this study that the anatomy or biomechanics of supraspinatus and infraspinatus tenotomy-induced rotator cuff injury in mice may not be fully extrapolated to human rotator cuff lesion with shoulder stiffness. Microsurgery-mediated supraspinatus injury in mice has been utilized to explain the cellular and molecular events underlying human shoulder disorders with rotator cuff tears [47,48]. Chronic inflammation and fibrosis are prominent features of this experimental rotator cuff injury model, which is consistent with excessive fibrotic tissue histopathology in human shoulder stiffness. In this study, the experimental animal model further explains the biological function of miR-29a indispensable in protecting against fibrosis in the development of shoulder stiffness. 

Taken together, a low miR-29a signaling escalates COL3A1, increasing fibrotic matrix formation in subacromial bursa in the development of rotator cuff lesion with shoulder stiffness. Gain of miR-29a function attenuates tendon tissue injury to attain shoulder function. Analysis offers a new molecular insight to rotator cuff tear with shoulder stiffness.

## 4. Materials and Methods

### 4.1. Patient Recruitment

All protocols for harvesting and examining human specimens were approved by Institutional Review Board (102-5462B, May 2013) of Kaohsiung Chang Gung Memorial Hospital. Written informed consent of all patients was obtained. Patients with rotator cuff tears with (*n* = 22) or without (*n* = 35) shoulder stiffness who had symptoms of impingement for over 3 months along with magnetic resonance imaging (MRI) of complete rotator cuff tear, as well as required surgery for open acromioplasty, lysis of adhesions, or rotator cuff repair were included in this study. Exclusion criteria of enrollment were patients who had the medical histories of shoulder disorders caused by traumatic fracture, surgery, instability, osteoarthritis, malignant disorders, hepatic or renal disorders, as previously described [6,7].

### 4.2. Diagnosis of Shoulder Stiffness

Passive range of motion (ROM) of shoulder was calculated upon measuring the maximum flexion and abduction angles of the involved shoulder in each patient in a sitting position using a goniometer. External and internal rotation angles of shoulder were examined as the arm was in a resting position. Non-stiffness was defined as ROM of shoulder with 180° forward flexion, 180° abduction, 90° external rotation, and 90° internal rotation. Shoulders with sum of ROM (SROM) <270° were categorized as stiff [6,7].

### 4.3. Harvest of Clinical Specimens

In the surgical lysis of adhesions or excision of adhesive subacromial synovium during rotator cuff repair, subacromial synovium specimens from greater tuberosity to coracoid within rotator cuff lesion were harvested for quantitative RT-PCR and immunohistochemistry. In a subset of the experiment, 10 mL venous blood was drawn from each patient preoperatively to probe serum miR-29 expression levels.

### 4.4. Histomorphometry, In Situ Hybridization, and Immunohistochemistry

Upon decalcifying in ethylenediaminetetraacetic acid/phosphate-buffered saline mixture, subacromial bursa specimens were embedded in paraffin and sectioned for hematoxylin-eosin staining for gross histology. Fibrotic tissue in specimens was assayed using Masson’s trichrome staining (Sigma-Aldrich, Co., St Louis, MO, USA), according to the manufacturer’s manual. Sections were analyzed microscopically using Zeiss microscope. Image acquisition was performed using Zeiss Image Analyzer. Subacromial bursa membrane thickness and fibrotic tissue area were quantified using the software of Zeiss Image Analyzer, according to the maker’s instructions. Six fields of 2 sections of each human specimen and 36 fields of 12 sections from 6 murine specimens were randomly selected for histomorphometry. Fibrosis was expressed as the percentile of area of fibrosis/total area of interest, as previously described [19]. For the detection of miR-29a transcripts in specimens, in situ hybridization was performed using miR-29a probes (Applied Biosystems, Carlsbad, CA, USA) labeled with digoxigenin along with digoxigenin antibody conjugated with horseradish peroxidase (Roche, Basel, Switzerland) [21]. In some experiments, collagen (COL)3A1 and α-smooth muscle actin (α-SMA) antibodies along with immunohistochemistry analysis kits containing IgG-conjugated with horseradish peroxidase (BioGenex, San Ramon, CA, USA) were utilized to probe fibrotic tissue and vessels in sections. Number of COL3A1 immunostained cells and α-SMA-immunostained vessels in 3 randomly selected fields in each section were counted.

### 4.5. RT-Quantitative PCR

Total RNA in specimens was extracted using Trizol reagent in a RNase-free condition. A total of 1 μg RNA was pipetted for reverse transcription into cDNA using ReadyScript^®®^ Two-Step cDNA Synthesis Kits (Sigma-Aldrich), according to the manufacturer’s instructions. In brief, 1 μg total RNA was mixed with 4 μL of ReadyScript cDNA Synthesis Mix with M-MLV reverse transcriptase, MgCl_2_, oligo-dT, and random primers. The mixtures were incubated in 25 °C for 5 min, 42 °C for 30 min and 85 °C for 5 min. For probing interleukin (IL)-1β, IL-6, IL-8, collagen (COL) A1, COL3A1, COL4A1, smooth muscle actin (SMA), disintegrin and metalloproteinase domain 12 (ADAM12), and 2-oxoglutarate 5-dioxygenase 2 (PLOD2), mRNA and housekeeping gene 18s rRNA, aliquots of cDNA were mixed with 2× TaqMan^®^ Universal PCR Master Mix (Applied Biosystems) along with specific primers -for PCR amplification using ABI 7900 Detection system (Applied Biosystems). The Ct value of each PCR amplification was computed automatically. Relative mRNA expression was calculated as 2^-ΔCt^, where ΔCt = Ct_gene_ − Ct_18s rRNA_. For probing miR-29 expression, 1 μg total RNA was reversely transcribed using TaqMan^®^ MicroRNA Reverse Transcription Kits (Thermo Fisher Scientific, Inc., Waltham, MA, USA) with 10× RT buffer, dNTP mix, RNAase inhibitor, and MultiScribe™ RT enzyme, according to the maker’s manuals. The RT mixtures were subjected to PCR amplification using 2× TaqMan^®^ Universal PCR Master Mix and specific primers for miR-29a, miR-29b, miR-29c, and housekeeping gene U6. Relative microRNA expression was calculated as 2^-ΔCt^, where ΔCt = Ct_miR-29a_ − Ct_U6_.

### 4.6. miR-29a Transgenic Mice

All protocols for animal use, surgery, and care were followed according to the animal wellbeing guidelines and approved by the Institutional Animal Care and Use Committee (IACUC No. 2015122306, December 2015) of Kaohsiung Chang Gung Memorial Hospital. Friend leukemia virus B (FVB) mice overexpressing miR-29a (miR-29aTg) driven by phosphoglycerate kinase (PGK) promoter were bred [19] and genotyped (forward: 5′-GAGGATCCCCTCAAGGATACCAAGGGAT-GAAT-3′; reverse: 5′-CTTCTAGAAGGAGTGTTTCTAGGTTCCGTCA-3′), as previously described [20]. Littermates that did not express the construct were utilized for wild-type mice.

### 4.7. Shoulder Rotator Cuff Lesion in Mice

Surgery-mediated rotator cuff injury in shoulders were performed, as previously described [23]. In brief, 8-week-old male wild-type or miR-29a transgenic mice were anesthetized using inhale isoflurane. After an aseptic incision of the skin of right forelimb, supraspinatus and infraspinatus tendon were exposed and cut off under a surgery microscope and followed by suturing the wound. Mice that received skin incision only were designated as sham controls. At 12 weeks postoperatively, animals were euthanatized using an overdose of anesthetics, and forelimbs were dissected for studies.

### 4.8. Desktop Digital Ultrasound Analysis

To assess the rotator cuff lesion in shoulders, animals were subjected to high-resolution ultrasound scanning (GE LOGIQ system, L 10-22 RS transducer). In brief, mice were anesthetized using inhale isoflurane and placed in a supine position at 6 and 12 weeks postoperatively. Sonographic images of the injured sites were captured according to the manufacturer’s manual. Fibrotic or hypertrophied tissue displayed high echogenicity.

### 4.9. In Vivo Near-Infrared Fluorescence Assay

Inflammation in the lesion site in the involved shoulders was traced using near-infrared fluorescence system, as previously described [24,25,26]. In brief, at 2 weeks postoperatively, mice were anesthetized using inhale isoflurane and injected 10 nM 2-deoxyglucose (LI-COR Bioscience, Lincoln, NE, USA) via tail vein. The uptake of fluorescence 2-deoxyglucose was scanned using near-infrared fluorescence in vivo imaging system (Pearl Impulse IRDye 800CW, LI-COR Bioscience, Lincoln, NE, USA). Fluorescence intensity of region of interest (5 mm × 5 mm) of lesion site was quantified using a 774-nm absorption wavelength of the system, according to the manufacturer’s instruction.

### 4.10. Gait Analysis

Gait profile and walking patterns of forelegs with injured shoulders in wild-type and miR-29aTg mice were analyzed using Catwalk system (Noldus Information Technology, Wageningen, The Netherlands). In brief, footprints of mice were recorded and computed by cameras and sensors of the system as animals walked through a walkway. Upon walking 5 times, standing time, max contact area, footprint length, footprint area, swing speed, and duty cycle of the involved right forelimbs (RF) were calculated and normalized with the uninjured left forelimbs (LF) using CatWalk software 9.1 and CatWalk XT’s Automatic Footprint Classification, according to the manufacturer’s instructions.

### 4.11. miR-29a Precursor Treatment

Fifteen μg expression vectors encoded miR-29a precursor along with 10 μg FIV-GAG and 5 μg VSVg plasmids were transfected into 293T cells. The transfected cells were further mixed with linker polybrene (Sigma-Aldrich) and followed by incubating in a 37 °C for 6 h. Lentivirus particles in medium were harvested upon ultrahigh-speed centrifugation, 10^8^ virus particles in 10 μL sterile saline were prepared for injecting into the injured site. Upon anesthesia, the injured shoulder in each animal was intraarticularly injected with 10 μL of lentivirus miR-29a precursor or lentivirus empty vector as mock under a flow guide of sonography at 1 week postoperatively.

### 4.12. Transfection of Human Tenocyte Cultures

Human tenocytes (Zen-Bio Inc., Research Triangle Park, NC, USA) were cultured in DMEM/F12 supplemented with 10% fetal bovine serum (FBS) and 1% penicillin/streptomycin (Gibco, Thermo Fisher Scientific Inc., Waltham, MA, USA). The 1 × 10^6^ cells were incubated in medium containing 1 ng/mL IL-1β, IL-6, and IL-8 (R&D Systems, Minneapolis, MN, USA) for 24 h. In a subset experiment, cells were transfected with 50 nM miR-29a precursor, antisense oligonucleotide, or scramble control (Applied Biosystems) using Lipofectamin™ 2000 (Invitrogen, Thermo Fisher Scientific, Inc., Waltham, MA, USA) and followed by incubating in 1 ng/mL IL-1β for 24 h.

### 4.13. Luciferase Reporter Assessment of miR-29a Binding to 3′-UTR of COL3A1

Wild-type (5′-UUCAAAAUGUCUCAAUGGUGCUA-3′) or 3-base pair mutant (5′-UUCAAAAUG UCUCAA*A*G*C*U*C*C*A*A-3′) of 3′-untranslated regions (3′-UTR) of COL3A1 were cloned and ligated into firefly luciferase pCRII-TORO vector. Aliquots of 10 ng pCRII-TORO vector along with 10 ng Renilla luciferase reporter (Promega Corporation, Madison, WI, USA) were transfected into 10^4^ tenocytes incubated in 96-well plates using Lipofectamin™ 2000 (Invitrogen, Thermo Fisher Scientific, Inc., Waltham, MA, USA). The luciferase reporter-transfected cells were further transfected with 30 nM miR-29a precursor, 30 nM antisense oligonucleotide, or 30 nM scrambled control and incubated in a 37 °C for 24 h. Luciferase activity in tenocytes were performed using Dual Luciferase Reporter Assay kits (BioVision, Milpitas, CA, USA), according to the manufacturer’s instructions. In brief, cells were harvested and mixed with Cell Lysis Buffer to isolate cell lysate. Aliquots of 50 μL cell lysates were mixed with 100 μL Substrate A and followed by mixing with 100 μL Substrate B. Luciferase activity in each well was quantified using luminometer and normalized with Renilla luciferase activity.

### 4.14. Statistical Analysis

Difference between human specimens in stiffness and non-stiffness groups was analyzed using the Wilcoxon test. Gender and number of involved shoulders in patients was analyzed by Chi-square test. Differences among sham-treated shoulder vs. surgery-injured shoulder in wild-type mice vs. miR-29a transgenic mice were verified by an ANOVA test and followed by a Bonferroni post-hoc test. *p* value < 0.05 was considered as statistical difference.

## Figures and Tables

**Figure 1 ijms-20-05742-f001:**
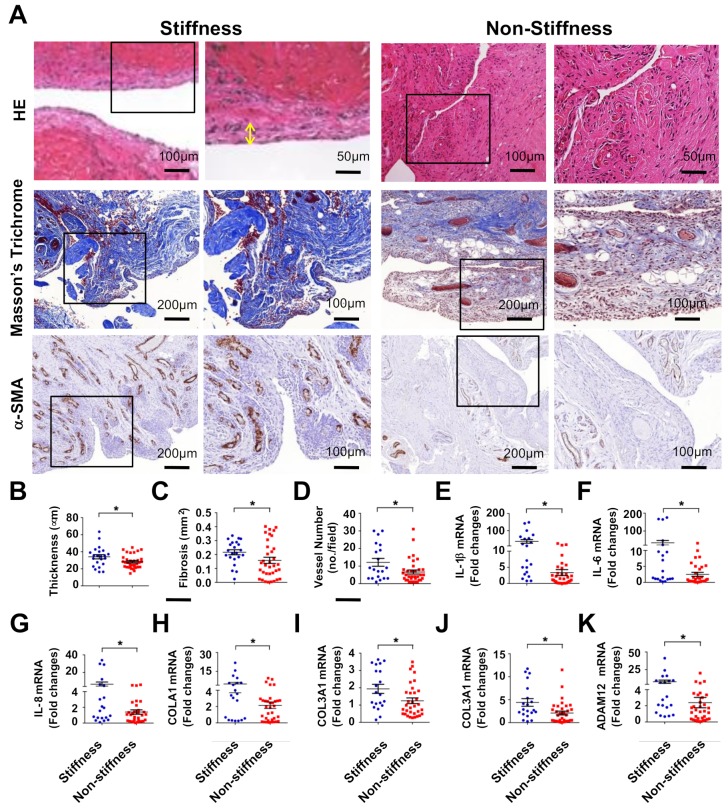
Analysis of fibrosis histopathology and fibrotic matrix expression of subacromial bursa in patients with rotator cuff lesion with or without stiffness. Subacromial bursa in the stiffness group show hypertrophy (yellow two-way arrow) as evident from hematoxylin and eosin (HE) staining. Masson’s trichrome staining displayed spacious fibrotic tissue (blue) rather than muscle tissue (red) in the stiffness group. Strong α-SMA immunostaining (brown) is exhibited in the stiffness group (**A**). The boxes stand for selected regions of interest for high power field images shown in panels 2 and 4 Subacromial bursa membrane thickness (**B**), fibrotic tissue area (**C**), and vessel formation (**D**) were significantly increased in the stiffness group. Inflammatory cytokines IL-1β (**E**), IL-6 (**F**), IL-8 (**G**) along with fibrotic matrices COL1A1 (**H**), COL13A1 (**I**), COL4A1 (**J**), and ADAM12 (**K**) expression were significantly upregulated in the stiffness group. Data are expressed as mean ± standard error of mean (SEM) from 22 patients in the stiffness group and 35 patients in the non-stiffness group. Asterisks (*) indicate significant differences between stiffness and non-stiffness group analyzed using Wilcoxon test (*p* < 0.05). HE, hematoxylin and eosin; α-SMA, α-smooth muscle actin; IL, interleukin; COL, collagen.

**Figure 2 ijms-20-05742-f002:**
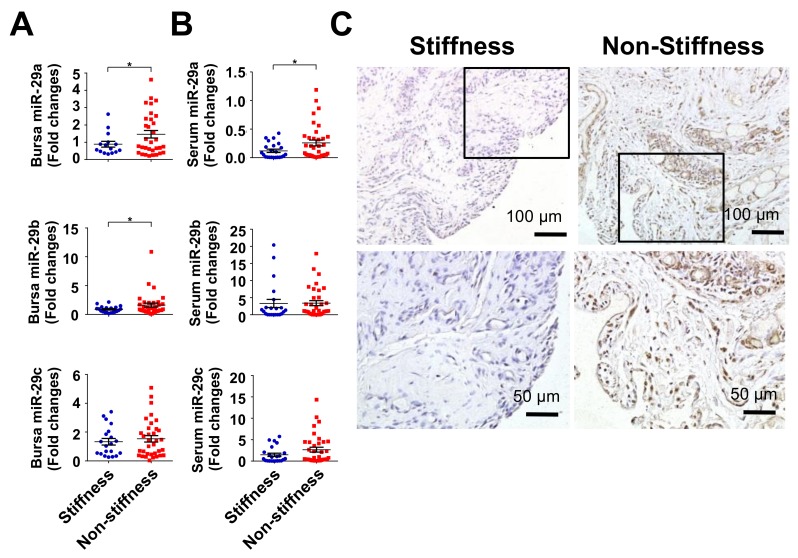
Analysis of miR-29 expression in subacromial bursa and serum. miR-29a and miR-29b expression were decreased in subacromial bursa in the stiffness group (**A**). Serum miR-29a rather than miR-29b and miR-29c levels were significantly downregulated in the stiffness group (**B**). Cells within subacromial bursa specimens in the stiffness group displayed weak miR-29a transcripts (brown) as compared to the non-stiffness group as evident from in situ hybridization (**C**). The boxes stand for selected regions of interest for high-power field images shown in right panels. Data are expressed as mean ± SEM from 22 patients with stiffness and 35 patients without stiffness. Asterisks (*) indicate significant differences between stiffness and non-stiffness group analyzed using Wilcoxon test (*p* < 0.05).

**Figure 3 ijms-20-05742-f003:**
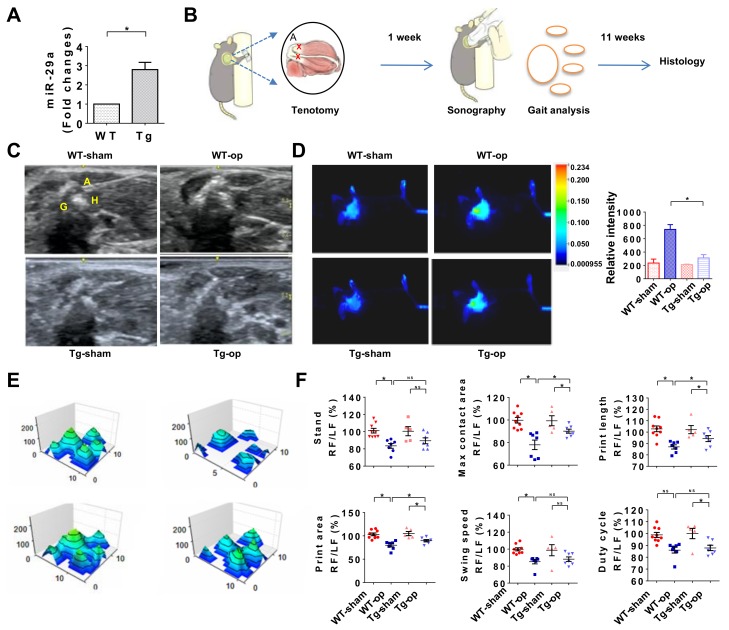
Analysis of rotator cuff lesion and gait profiles of wild-type mice and miR-29aTg mice. miR-29a expression was significantly increased in miR-29aTg mice (**A**). Schematic drawings for surgery-induced rotator cuff injury in mice (**B**). Echogenicity of injured shoulders was compromised in miR-29aTg mice (**C**). Inflammation in the injured tendon as evident from fluorescence 2-deoxyglucose uptake was downregulated in miR-29aTg mice (**D**). The tenotomy-mediated irregular footprint histograms (**E**) along with decreased stand time, maximum contact area, footprint length, footprint area, swing speed, and duty cycle (**F**) of forelegs with injured shoulders were improved in miR-29aTg mice. Data are expressed as mean ± SEM from 6–9 mice analyzed using ANOVA test and Bonferroni post-hoc test. Asterisks * (*p* < 0.05) indicate significant difference between groups. A, acromion; G, glenoid; H, humeral head; Op, tenotomy surgery; RF, right forelimb; LF, left forelimb; NS, not significant.

**Figure 4 ijms-20-05742-f004:**
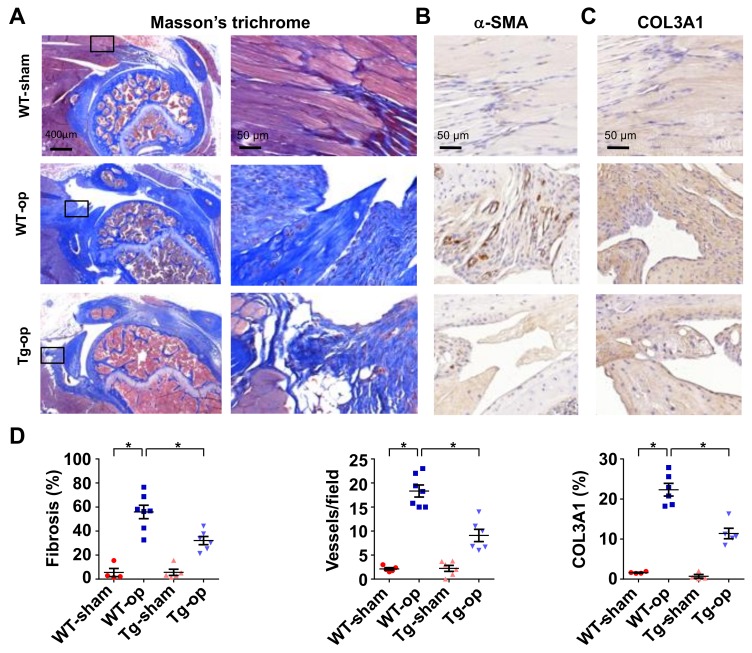
Histological analysis of rotator cuff injury in wild-type mice and miR-29aTg mice. Injured tendon in miR-29aTg mice showed moderate response to the tenotomy-mediated fibrotic tissue formation (blue) as evident from Masson’s trichrome staining (**A**). Boxes stand for selected regions of interest for high-power field images shown in right panels. The injured sites in miR-29aTg mice showed weak α-SMA immunostaining (**B**) and COL3A1 immunostaining (brown) (**C**). miR-29a overexpression significantly improved fibrotic tissue area, vessel number and COL3A1 production (**D**). Data are expressed as mean ± SEM from 5–7 mice analyzed using ANOVA test and Bonferroni post-hoc test. Asterisks * indicate significant difference between groups (*p* < 0.05).

**Figure 5 ijms-20-05742-f005:**
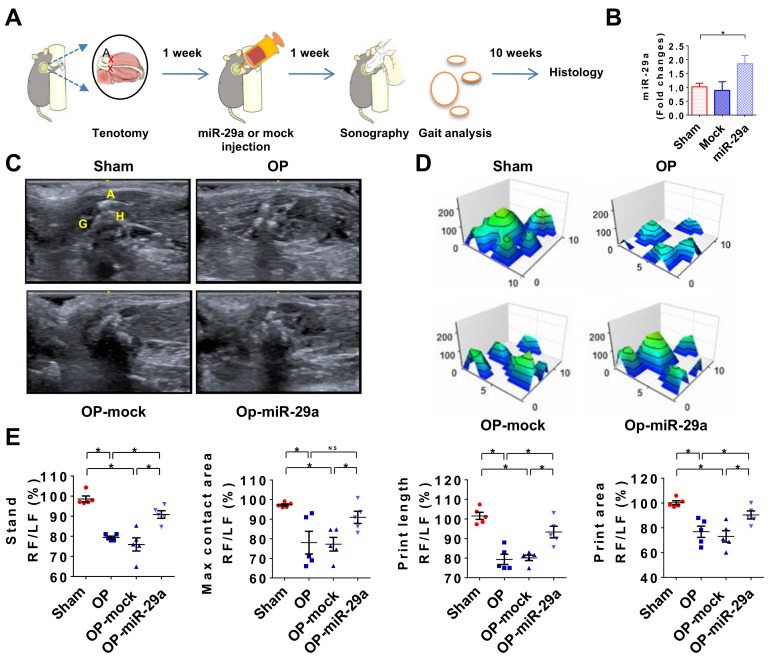
miR-29a precursor treatment improvement of shoulder injury. Schematic drawings for miR-29a precursor treatment for shoulder injury (**A**). miR-29a expression was increased at 1 week upon miR-29a precursor treatment (**B**). Echogenicity of injured shoulder was compromised upon miR-29a precursor treatment (**C**). Tenotomy-mediated irregular footprint (**D**) along with stand time, maximum contact area, footprint length, and footprint area (**E**) were improved upon miR-29a precursor treatment. Data are expressed as mean ± SEM from 5–6 mice. Asterisks * (*p* < 0.05) indicate significant difference between groups analyzed using ANOVA test and Bonferroni post-hoc test. A, acromion; G, glenoid; H, humeral head; Op, tenotomy surgery; RF, right forelimb; LF, left forelimb; NS, not significant.

**Figure 6 ijms-20-05742-f006:**
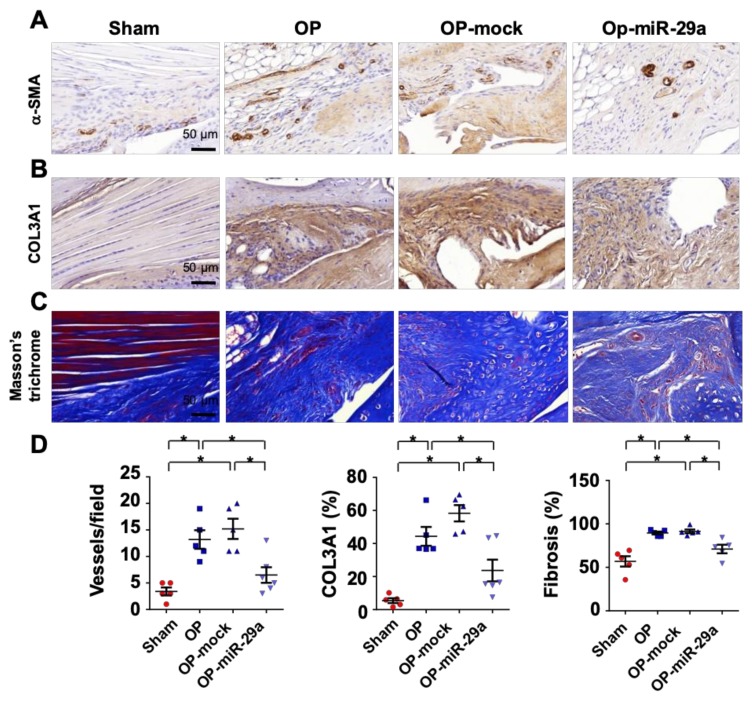
Histopathological analysis of injured shoulders with or without miR-29a precursor treatment. Injured sites in the miR-29a-treated group showed moderate α-SMA immunostaining (brown) (**A**) and COL3A1 immunoreaction (brown) (**B**) along with mild fibrosis tissue formation (blue) as evident from Masson’s trichrome staining (**C**). The tenotomy upregulation of vessel formation, COL3A1 production and fibrotic tissue area were compromised upon miR-29a precursor treatment (**D**). Data are expressed as mean ± SEM from 5–6 mice analyzed using ANOVA test and Bonferroni post-hoc test. Asterisks * indicate significant difference between groups (*p* < 0.05).

**Figure 7 ijms-20-05742-f007:**
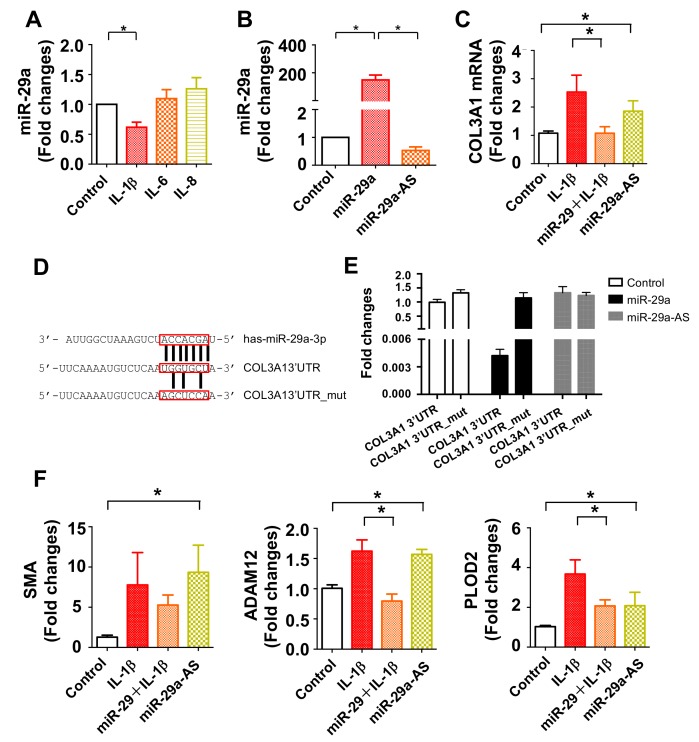
Analysis of miR-29a action to human tenocytes. IL-1β rather than IL-6 and IL-8 significantly decreased miR-29a expression in tenocytes. Cell cultures were incubated in 1 ng/mL IL-1β, IL-6, or IL-8 for 24 h (**A**). miR-29a expression was increased in tenocytes upon miR-29a precursor transfection, whereas miR-29a antisense oligonucleotide (miR-29a-AS) transfection reduced the expression (**B**). miR-29 precursor transfection attenuated the IL-1β-mediated COL3A1 expression. miR-29a knockdown alone provoked COL3A1 expression (**C**). Sequence (red boxes) of the miR-29a targeted 3′-untranalated region (3′-UTR) and 3-base pair mutant of COL3A1 were constructed for luciferase reporter assay (**D**). miR-29a precursor transfection decreased luciferase reporter activity of 3′-UTR rather than mutant of COL3A1 in tenocytes, whereas miR-29a-AS transfer increased the luciferase reporter activity (**E**). miR-29a precursor attenuated the IL-1b-augmented SMA, ADAM12, and PLOD2 expression. miR-29a-AS increased fibrosis marker expression in tenocytes (**F**). Data are expressed as mean ± SEM calculated from three experiments analyzed using ANOVA test and Bonferroni post-hoc test. Asterisks * indicate significant difference between groups (*p* < 0.05).

**Table 1 ijms-20-05742-t001:** Demography of patients with rotator cuff tear with or without shoulder stiffness.

	Stiffness	Non-Stiffness	*p* Value
Participants	22	35	
Gender			
Female	19 (86.4%)	24 (68.6%)	0.8
Male	3 (13.6%)	11 (31.4%)	-
Involved shoulder			
Right	13 (59%)	24 (68.6%)	0.78
Left	9 (41%)	11 (31.4%)	-
Age (years)	62.1 ± 2	64.8 ± 1.2	0.12
	(48–77)	(52–80)	
BMI	25.1 ± 1	25.4 ± 0.5	0.42
	(16.6–32.2)	(20.8–33.4)	
Functional score of Constant and Murley	39.2 ± 2.2	63.5 ± 1.5	<0.001
	(21–67)	(49–85)	
Sum of passive range of motion (SROM)	196.6 ± 10.1	365 ± 9.7	<0.001
	(115–270)	(275–520)	
Flexion	83.4 ± 5	147.1 ± 3.5	<0.001
	(35–130)	(100–180)	
Abduction	72.5 ± 4.8	119.7 ± 3.5	<0.001
	(40–110)	(75–165)	
External rotation	15.2 ± 2.6	41.9 ± 3.1	<0.001
	(0–40)	(10–75)	
Internal rotation	26 ± 6	54.6 ± 4.6	<0.001
	(0–80)	(0–90)

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
