# Peer review of "MicroRNA-29a Mitigates Subacromial Bursa Fibrosis in Rotator Cuff Lesion with Shoulder Stiffness"

_ijms, 2019, doi:10.3390/ijms20225742_

Round 1

Reviewer 1 Report

The authors made some rewriting changes, but to the comments they did not reply properly. I'll not be able to give a positive review on the revised version.

In brief, if a reviewer mention that an experiment is not performed well, it is not enough to change the text. The authors should also make wet lab experiments. If a reviewer indicates that images do not show what the authors write, the authors should take new images or remove the text/image. Changing the text would not make a difference.

Last the authors wrote few times that the revised version was read by a native English speaker. Regretfully, many mistakes are to be found, for example: The second sentence in the results (section 2.1) , and the last sentence in this same paragraph.

Author Response

The authors made some rewriting changes, but to the comments they did not reply properly. I’ll not be able to give a positive review on the revised version.

In brief, if a reviewer mention that an experiment is not performed well, it is not enough to change the text. The authors should also make wet lab experiments. If a reviewer indicates that images do not show what the authors write, the authors should take new images or remove the text/image. Changing the text would not make a difference.

Authors’ responses: Thank you for this comment. We provided additional evidence of miR-29a downregulation of fibrosis marker expression, like smooth muscle actin (SMA), disintegrin and metalloproteinase domain 12 (ADAM12) and 2-oxoglutarate 5-dioxygenase 2 (PLOD2), in inflamed tenocytes in the revised version (Lines 199-202), which now read as follows:

miR-29a precursor also attenuated fibrotic marker expression, like smooth muscle actin (SMA), disintegrin and metalloproteinase domain 12 (ADAM12) and 2-oxoglutarate 5-dioxygenase 2 (PLOD2), in inflamed tenocytes (Fig. 7F). miR-29a-AS significantly augmented SMA, ADAM12 and PLOD2 expression in tenocytes without IL-1β stress (Fig. 7F).

Last the authors wrote few times that the revised version was ready for a native English speaker. Regretfully, many mistakes are to be found, for example: The second sentence in the results (section 2.1), and the sentence in the same paragraph.

Authors’ responses: We apologize for the typos and misspelling so that make you spend more time to figure it out. We re-wrote the text and corrected typos in the revised version. We also re-wrote some protocols in a more detailed way.

Reviewer 2 Report

All the comments have been addressed by the authors.

However, it is necessary to improve the English language of the newly written parts since several orthographic mistakes show up and some sentences are not completely clear.

Concerning the methods section some parts should still be described in more detail, e.g. the Luciferase Reporter Assessment.

In addition, anesthesia of mice should be described for Ultrasound analysis.

Author Response

All the comments have been addressed by the authors. However, it is necessary to improve the English language of the newly written parts since several orthographic mistakes show up and some sentences are not completely clear.

Authors’ responses: We apologize for the typos and misspelling so that make you to spend more time to figure it out. We re-wrote the text and corrected typos in the revised version.

Concerning the methods section some parts should still be described in more detail, e.g. the Luciferase Reporter Assessment.

Authors’ responses: Thank you for this comment. We rewrote the protocol in the revised version (Lines 387-397), which now read as follows:

Wild-type (5’-UUCAAAAUGUCUCAAUGGUGCUA-3’) or 3-base pair mutant (5’-UUCAAAAUGUCUCAAAGCUCCAA-3’) of 3’-untranslated regions (3’-UTR) of COL31 were cloned and ligated into firefly luciferase pCRII-TORO vector. Aliquots of 10 ng pCRII-TORO vector along with 10 ng Renilla luciferase reporter (Promega Corporation, WI, USA) were transfected into 104 tenocytes incubated in 96-well plates using Lipofectamin™ 2000 (Invitrogen). The luciferase reporter-transfected cells were further transfected with 30 nM miR-29a precursor, 30 nM antisense oligonucleotide or 30 nM scrambled control and incubated in a 37 ℃ for 24 hours. Luciferase activity in tenocytes were performed using Dual Luciferase Reporter Assay kits (BioVision, CA), according to the manufacturer’s instructions. In brief, cells were harvested and mixed with Cell Lysis Buffer to isolate cell lysate. Aliquots of 50 μl cell lysates were mixed with 100 μl Substrate A and followed by mixing with 100 μl Substrate B. Luciferase activity in each well was quantified using luminometer and normalized with renilla luciferase activity.

In addition, anesthesia of mice should be described for Ultrasound analysis.

Authors’ responses: We re-wrote the sentences in the revised version, now read as follows:

To assess the rotator cuff lesion in shoulders, animals were subjected to high-resolution ultrasound scanning (GE LOGIQ system, L 10-22 RS transducer). In brief, mice were anesthetized using inhale isoflurane and placed in a supine position at 6 and 12 weeks postoperatively. Sonographic images of the injured sites were captured, according to the manufacturer’s manual. Fibrotic or hypertrophied tissue displayed high echogenicity.

This manuscript is a resubmission of an earlier submission. The following is a list of the peer review reports and author responses from that submission.

Round 1

Reviewer 1 Report

The manuscript entitled "MicroRNA-29a Mitigates Subacromial Bursa Fibrosis in Rotator Cuff Lesion with Shoulder Stiffness" describes miR-29a as a potential novel cure to help patients with shoulder stiffness by reducing fibrosis through reducing Col3a1. This effect was demonstrated in a mouse model and by in vitro experiments with human tendocytes.

However, already in the introduction I have the feeling that the authors do not clearly understand the mechanism of action of microRNAs (see comments for the abstract and the introduction).

In addition, I have some concerns about the presentation of the results. Especially, the figure legends are much to short. You can’t follow the results by reading them.

The discussion is often redundant in repeating the same again and again. And finally, the methods are not described in enough detail (please see comments).

comments:

Abstract

line 22: it should be mentioned that the miR-29a transgenic mice are overexpressing miR29a

line 22: regular gait not regulargait

line 28: miRs are not targeting transcription; they mainly work posttranscriptional; maybe it is better to write something like: “directly targeting COL3a1 mRNA levels”

line 28: I think it is not correct to designate microRNA actions as epigenetic mechanisms (epigenetic mechanisms are for example DNA methylations or histone modifications). Please change that

Introduction

line 41: “to regulate gene expression post-transcriptional”.

line 42: microRNAs do not work post-translational. They do not modify proteins. Please delete that.

Results

line 89: I think you mean miR-29 family members

line 90: the same here: the miR-29 family

line 94: you didn’t prove that the cells stained for miR29a are fibroblasts. So you can’t write that here.

line 107: gait profile not gat profile

line 130: as already mentioned above you can’t tell that fibroblasts around tendons strongly displayed the vessel marker aSMA because you didn’t counterstain them with a second marker.

line 142/143: was the miR-29a precursor treatment performed in wild type mice?? Please indicate that.

line 144: the description of the experiment here does not mirror the scheme in Figure 5A. Please figure out if tenotomy or lentivirus injection was done first.

line 145/146: this statement does not mirror Fig 5B. Please clarify that issue. Do you mean the sham group as Ctr group??

line 173/174: were the cells for the miR29a silencing group also treated with IL-1B. Please clarify.

Discussion

line 195: again: you can’t know that the cells are fibroblasts. You have to prove that.

line 218: miR-29 family

line 219: miR-29a expression

line 223: miR-29aTg mice.

line 229: again the authors talk about fibroblasts what they can’t know.

Figures: The presentation of results as well as their description (figure legends) has to be improved significantly for publication.

Figure 1: Please describe the color code for HE (A) and Trichrome (A) stainings since not every reader might know these methods. The same for the immunostaining of SMA.

Figure 2: Please describe the color code for in situ hybridization. (C) You should describe what the arrows mean. And how do you know that the cells that are stained are fibroblasts? There is no counterstaining for such a fibroblast marker. So you can’t claim that the cells are fibroblasts.

Figure 3: (D) I don’t see a big difference between the picture for WT-op and TG-op. However, please label the bars in the graph. There are four bars, but only three pictures?  

(F) The legend says that 6 mice were analyzed in each group. However, for the WT sham group there are more than 6 dots in each diagram.

Figure 4: Please describe the color code for Trichrome (A) stainings since not every reader might know these methods. The same for the immunostaining of SMA and COL3a1.

(D): Graphs should be presented bigger. Dots are hardly recognizable.

Figure 5: (D): Graphs should be presented bigger. Dots are hardly recognizable. Moreover it would be better to use a new letter for the bar graphs (E).

Figure 6: Again: Please describe the color code for Trichrome (C) stainings and immunostainings of SMA (A) and COL3a1 (B). Graphs should be presented bigger. Dots are hardly recognizable.

Figure 7: Figure Legend should describe in more detail what was done in the experiment. Especially for (C) it is not exactly clear what was done with the cells.

Methods:

The methods have to be presented in more detail. Devices and software that were used should be mentioned for the experiments etc

Especially miR29a precursor treatment (line 316ff) has to be described in much more detail.

Author Response

Response to Reviewer #1

The manuscript entitled "MicroRNA-29a Mitigates Subacromial Bursa Fibrosis in Rotator Cuff Lesion with Shoulder Stiffness" describes miR-29a as a potential novel cure to help patients with shoulder stiffness by reducing fibrosis through reducing Col3α1. This effect was demonstrated in a mouse model and by in vitro experiments with human tendocytes.

However, already in the introduction I have the feeling that the authors do not clearly understand the mechanism of action of microRNAs (see comments for the abstract and the introduction).

In addition, I have some concerns about the presentation of the results. Especially, the figure legends are much too short. You can’t follow the results by reading them.

The discussion is often redundant in repeating the same again and again. And finally, the methods are not described in enough detail (please see comments).

Author response: We apologize for not interpreting the text in a more detailed way so that make you spending more time to figure out it. We re-wrote the text, especially the methodology, discussion and figure legends in the revised version.

comments:

Abstract
line 22: it should be mentioned that the miR-29a transgenic mice are overexpressing miR29a
line 22: regular gait not regulargait
line 28: miRs are not targeting transcription; they mainly work posttranscriptional; maybe it is better to write something like: “directly targeting COL3α1 mRNA levels”

Author response: We corrected the typos throughout the text. In addition, we re-wrote the sentences (Lines 42-43) in the revised version, which now read as follows:

MicroRNAs, small non-coding RNA with around 22 nucleotides, are shown to regulate gene expression post-transcriptionally during tissue development and damage [8, 9].

line 28: I think it is not correct to designate microRNA actions as epigenetic mechanisms (epigenetic mechanisms are for example DNA methylations or histone modifications). Please change that

Author response: Thank you for this comment. We changed “epigenetic mechanistic” to “molecular mechanism” (Line 29) in the revised version.

Introduction
line 41: “to regulate gene expression post-transcriptional”.
line 42: microRNAs do not work post-translational. They do not modify proteins. Please delete that.

Author response: we re-wrote the sentences (Lines 42-43) in the revised version, which now read as follows:

MicroRNAs, small non-coding RNA with around 22 nucleotides, are shown to regulate gene expression post-transcriptionally during tissue development and damage [8, 9].

Results
line 89: I think you mean miR-29 family members
line 90: the same here: the miR-29 family
line 107: gait profile not gat profile

Author response: We revised “the miR-29a family” to “the miR-29a family” and correct “gat profile” to “gait profile” throughout the text in the revised version.

line 94: you didn’t prove that the cells stained for miR29a are fibroblasts. So you can’t write that here.
line 130: as already mentioned above you can’t tell that fibroblasts around tendons strongly displayed the vessel marker αSMA because you didn’t counterstain them with a second marker.

Author response: Thank you, point taken. We revised “subacromial bursa fibroblasts” into “subacromial bursa tissue” throughout the text and figure legends.

line 142/143: was the miR-29a precursor treatment performed in wild type mice?? Please indicate that.

Author response: Yes. Wild-type mice with rotator cuff injury were injected with lentivirus miR-29a precursor. We re-wrote the sentences (Lines 160-161) in the revised version, now read as follows:

Lentivirus human miR-29a precursor or lentivirus empty vector as mock was injected into the injured site in wild-type mice under sonography guide at 1 week postoperatively (Fig. 5A).

line 144: the description of the experiment here does not mirror the scheme in Figure 5A. Please figure out if tenotomy or lentivirus injection was done first.

Author response: Thank you for bringing this to our awareness. Lentivirus miR-29a precursor was injected at 1 week postoperatively. We rewrote the sentences (Lines 160-161) and re-drew the scheme in the revised version.

line 145/146: this statement does not mirror Fig 5B. Please clarify that issue. Do you mean the sham group as Ctr group??

Author response: In Fig. 5, sham stands for mice received skin incision only were designated as sham controls. Mock resembles mice with rotator cuff injury received lentivirus empty vector only rather than lentivirus miR-29a precursor. We rewrote the methodology (Lines 367-369) in the revised version, now read as follows:

108 virus particles in 10 μl sterile saline were prepared for intra-articular injection into the injure site. Upon anesthesia, the injured shoulder in each animal was intra-articularly injected with lentivirus miR-29a precursor or lentivirus empty vector as mock under a flow guide of sonography.

line 173/174: were the cells for the miR29a silencing group also treated with IL-1β. Please clarify.

Author response: In Fig 7B, 7C and 7D, the miR-29a-AS group stands for miR-29a antisense oligonucleotide transfected human tenocytes without IL-1β stress. We rewrote the sentences (Lines 195-196) in revised version, now read as follows:
miR-29a antisense oligonucleotide (miR-29a-AS) transfection decreased miR-29a resulted in an increase in COL3A1 expression in tenocytes without IL-1β stress.

Discussion
line 195: again: you can’t know that the cells are fibroblasts. You have to prove that.

Author response: We revised “subacromia bursa fibroblasts” into “subacromia bursa tissue” throughout the text and figure legends.

line 221: miR-29 family
line 222: miR-29a expression
line 226: miR-29aTg mice.
line 229: again the authors talk about fibroblasts what they can’t know.

Author response: We revised “the miR-29a family” to “the miR-29a family” and correct “gat profile” to “gait profile” throughout the text in the revised version.
.
Figures: The presentation of results as well as their description (figure legends) has to be improved significantly for publication.

Figure 1: Please describe the color code for HE (A) and Trichrome (A) stainings since not every reader might know these methods. The same for the immunostaining of SMA.

Author response: We rewrote the legend of Fig. 1 (Lines 88-97) in the revised version, which now read as follows:

Analysis of fibrosis histopathology and fibrotic factor expression of subacromial bursa in patients with rotator cuff lesion with or without stiffness. Subacromial bursa in the stiffness group show hypertrophy as evident from hematoxylin and eosin (HE) staining. Masson’s trichrome staining display over-development of fibrotic tissue (blue) rather than muscle tissue (red) in the stiffness group. Strong α-SMA immunostaining (brown) exhibit in the stiffness group (A). Subacromial bursa membrane thickness (B), fibrotic tissue area (C) and vessel formation (D) were significantly increased in the stiffness group. Fibrogenic factors IL-1β (E), IL-6 (F) IL-8 (G) along with fibrotic matricies COL11 (H), COL131 (I), COL41 (J) and ADAM12 (K) expresion were significantly upregulated in the stiffness group. Data are expressed as mean ± SEM from 22 patients in the stiffness group and 35 patients in the non-stiffness group. Asterisks (*) indicate significant differences between stiffness and non-stiffness group analyzed using Wilcoxon test (P < 0.05).

Figure 2: Please describe the color code for in situ hybridization. (C) You should describe what the arrows mean. And how do you know that the cells that are stained are fibroblasts? There is no counterstaining for such a fibroblast marker. So you can’t claim that the cells are fibroblasts.

Author response: We rewrote the legend of Fig. 2 (Lines 108-114) in the revised version, which now read as follows:
Analysis of miR-29 expression in subacromial bursa and serum. miR-29a and miR-29b expression were decreased in subacromia bursa specimens of the stiffness group (A). Serum miR-29a rather than miR-29b and miR-29c levels were significantly downregulated in the stiffness group (B). Cells within subacromial bursa specimens in the stiffness group display weak miR-29a transcripts (brown) as compared to the non-stiffness group as evident from in situ hybridization (C). Data are expressed as mean ± SEM from 22 patients with stiffness and 35 patients without stiffness. Asterisks (*) represents exit of significant different between stiffness and non-stiffness group analyzed using Wilcoxon test (P<0.05).

Figure 3: (D) I don’t see a big difference between the picture for WT-op and TG-op. However, please label the bars in the graph. There are four bars, but only three pictures? (F) The legend says that 6 mice were analyzed in each group. However, for the WT sham group there are more than 6 dots in each diagram.

Author response: We provided near infrared fluorescence image of WT-Sham group and rewrote the legend of Fig. 3 (Lines 131- 140) in the revised version, which now read as follows:

Analysis of rotator cuff lesion and gait profile of wild-type mice and miR-29a transgenic mice. miR-29a expression was significantly increased in miR-29aTg mice (A). Schematic drawings of creation and assessment of surgery-induced rotator cuff injury in mice (B). Echogenicity of injured shoulder was compromised in miR-29aTg mice (C). Inflammation in the injured site as evident from fluorescence 2-deoxyglucose uptake was downregulated in miR-29aTg mice (D). Irregular footprint histograms (E) along with decreased stand time, maximum contact area, footprint length, footprint area, swing speed, and duty cycle (F) of forelegs with involved shoulders were improved in miR-29aTg mice. Data are expressed as mean ± SEM from 6 - 9 mice analyzed using ANOVA test and Bonferroni post hoc test. Asterisks * (P < 0.05) and ** (P < 0.01) indicate significant different between groups. A, acromion; G, glenoid; H, humeral head; Op, tenotomy surgery; RF, right forelimb; LF, left forelimb.

Figure 4: Please describe the color code for Trichrome (A) stainings since not every reader might know these methods. The same for the immunostaining of SMA and COL3a1. (D): Graphs should be presented bigger. Dots are hardly recognizable.

Author response: We re-wrote the legend of Fig. 4 (Lines 149 – 155) in the revised version, which now read as follows:

Histological analysis of rotator cuff injury in wild-type mice and miR-29aTg mice. Injured sites in miR-29aTg mice showed a moderate response to the tenotomy-mediated fibrotic tissue formation (blue) as evident from Masson’s trichrome staining (A). The tenotomy-upregulated vessel formation as evident from vessel marker α-SMA immunostaining (B) and fibrotic matrix COL31 immunostaining (brown) (C) were reduced in miR-29aTg mice. miR-29a overexpression significantly improved fibrotic tissue area, vessle number and COL3A1 production (D). Data are expressed as mean ± SEM from 5 - 7 mice analyzed using ANOVA test and Bonferroni post hoc test. Asterisks * indicate significant different between groups (P < 0.05).

Figure 5: (D): Graphs should be presented bigger. Dots are hardly recognizable. Moreover, it would be better to use a new letter for the bar graphs (E).

Author response: We revised the size of Fig. 5 and added a new letter for Fig. 5E in the revised version.

Figure 6: Again: Please describe the color code for Trichrome (C) stainings and immunostainings of SMA (A) and COL3a1 (B). Graphs should be presented bigger. Dots are hardly recognizable.

Author response: We revised the size of Fig. 6 and re-wrote the legend (Lines 181 – 187) in the revised version, which now read as follows:

Histopathological analysis of injured shoulder with or without miR-29a precursor treatment. Injured sites in the miR-29a-treated group showed moderate immunostaining of vessel marker -SMA (brown) (A) and fibrotic matrix COL31 (brown) (B) along with mild fibrosis tissue formation (blue) as evident from Masson’s trichrome staining (C). The tenotomy upregulation of vessel formation, collagen 31 production and fibrotic tissue area were compromised upon miR-29a precursor treatment. Data are expressed as mean ± SEM from 5 - 6 mice analyzed using ANOVA test and Bonferroni post hoc test. Asterisks * indicate significant different between groups (P < 0.05).

Figure 7: Figure Legend should describe in more detail what was done in the experiment. Especially for (C) it is not exactly clear what was done with the cells.

Author response: We rewrote the legend of Fig. 7 (Lines 202 – 212) in the revised version, which now read as follows:
Analysis of miR-29a action to human tenocyte cultures. IL-1β rather than IL-6 and IL-8 significantly Decreased miR-29a expression in tenocytes. Cell cultures were incubated in 1 ng/ml IL-1β, IL-6 or IL-8 for 24 hours (A). miR-29a expression was increased in cells upon miR-29a precursor transfection, whereas miR-29a antisense oligonucleotide (miR-29a-AS) transfection reduced the expression (B). miR-29 precursor transfection attenuated the IL-1β-mediated COL3A1 expression, whereas miR-29a knockdown alone provoked COL3A1 expression (C). Sequence of the miR-29a targeted 3’-untranalated region (3’-UTR) and 3-base pair mutant of COL3A1 were designed and constructed for luciferase reporter assay. miR-29a precursor transfection decreased luciferase reporter activity of 3’-UTR rather than mutant of COL3A1 in tenocytes, whereas miR-29a-AS transfer increased the luciferase reporter activity (D). Data are expressed as mean ± SEM calculated from 3 experiments analyzed using ANOVA test and Bonferroni post hoc test. Asterisks * indicate significant different between groups (P < 0.05).

Methods:

The methods have to be presented in more detail. Devices and software that were used should be mentioned for the experiments, etc.
Especially miR29a precursor treatment (line 316ff) has to be described in much more detail.

Author response: We rewrote the methodology in the revised version. For example, the methodology of miR-29a precursor treatment (Lines 364 - 370) were written as follows:
15 μg expression vectors encoded miR-29a precursor along with 10 μg FIV-GAG and 5 μg VSVg plasmids were transfected into 293T cells. The transfected cells were further mixed with linker polybrene (Sigma-Aldrich) and followed by incubating in a 37 °C for 6 hours. Lentivirus particles in medium were harvested upon ultrahigh speed centrifugation, 108 virus particles in 10 μl sterile saline were prepared for intra-articular injection into the injure site. Upon anesthesia, the injured shoulder in each animal was intra-articularly injected with 10 μl of lentivirus miR-29a precursor or lentivirus empty vector as mock under a flow guide of sonography at 1 week postoperatively.

Reviewer 2 Report

In the manuscript entitled: MicroRNA-29a Mitigates Subacromial Bursa Fibrosis inRotator Cuff Lesion with Shoulder Stiffness Ko at al., sudy the functional involvement of miR-29a in tendocytes and fibrosis and relate their observation to decrease miR-29a in rotator cull tissue from patients with shoulder stiffness and controls. The study is overall interesting and relevant to the field. It is also corroborates to other studies showing the involvement of miR-29a in tissue fibrosis and other pathologies. Moreover, their study reinforce a therapeutic potential for miR-29a.

However, there are several issues/ critics that must be addressed before publication, which are listed here under.

General comments:

Shoulder diseases are categorized between chronic and acute. The chronic is age associated and as such tissue histology differs from the acute condition. The author should specify on the condition they have studied. Moreover, shoulder diseases are different pathologies, the authors should be more specific to which pathology they studied. The tissue that was used in human is not clear, was it the shoulder impingement, which part of the tissue was used for molecular analysis. More details will help the reader. Since it is a highly heterogenic tissue, it can explain the huge variations between individuals that are found in the molecular analysis. The human shoulder anatomy and biomechanics are not to be found in a mouse or any animal walking on four legs. This should be changed throughout the paper. Therefore, the results in mouse show an effect of mir-29a in fibrotic tissue. Moreover, what is the tissue that was studied in mouse: muscle tendon. The histology differs from that in human. How quantification from histological images were made is not entirely clear. In principles, a colorimetric signal cannot be quantified. Trichome staining is not quantifiable. The authors write: Line 272: Number of cells positive for COL3a1 immunostaining and number of vascular tissue positive for a-SMA in 3 randomly selected fields in each section were counted.” The resolution in Fig. 2 and Fig 4 of those staining is insufficient for counting. Moreover, COL3a1 is secreted, who it is possible to count positive cells? As a note: Quantification of collagen fluorescence signal is much more reliable. In addition, fibrosis in the RC is associated with an increase in fat (detected by MRI) and an increase in fatty droplets (Nile Red staining), as signal that can also be quantified.

Specific comments:

Results and methods

Fig. 1 and 2 the variations are enormous, was a test for outliers was applied?

Fig. 1A, for the SMA staining, a different contrast and brightness was applied.

The statistical test should be specified in the legends not only in the results.

The presentation of the in situ hybridization results is not convincing. We also know that miRs are secreted and since they are part of cytoplasmic RNAdecay signal other than nuclear was reported. Alternatively, images should be taken using a higher magnification.

The methods for the RTqPCR are not entirely clear and crucial details are missing in order to assess the results: how cDNA was made with oligo d(T) and/or random primer? Normalization to the 18S rRNA is often considered unreliable as the CT values are very low compared with mRNAs. The authors should show CT values. Moreover, it cannot be used for miRNA normalization. This requires another (unchanged) miRNA in combination with spike. At minimum the authors should show CT values and an unrelated miRNA with unaltered levels.

The in vivo near infrared fluorescence assay is not commonly used and could be elaborated in more details, also in the Results section. The authors refer to two different references, this is confusing.

Fig. 6 the effect on tissue histology is not very convincing: a treatment with miR0-29a induces fatty formation.

The legend to Fig. 7 is mixed up or incomplete. The color-code of the bars is not specify. What are three repeated experiments? What is “forced expression” is it overexpression? Is this overexpression under physiological conditions? It is essential to show the expression of other genes (also house-keeping) in this overexpression condition. Mir-29a has over 1000 predicted target genes, in mirDB, assessing one gene is not sufficient to make a solid point. Exp. 7D should be explained in more details, the third bar (miR-29a +col3A1-3’UTR) is not to be seen, also not SD.

Discussion:

As the study shows the effect of miR-29a on fibrosis and Col1A expression, the discussion should focus on that. The relevance to shoulder disease is important, but indirect.

Nearly the whole first section of the Discussion is Background information and should be placed in the Introduction.

Grammar:

The manuscript contains unclear terms and unclear sentences. Few are listed here under. But it should carefully checked.

Line 37: shoulder disorders (not one disorder).

Line 41 should be “are”.

Lines 202-203, not coherent. The following sentence also does not make sense.

miRNA affect mRNA delay and hence mRNA levels. Gene expression is not a specific term.

“Fibrotic reactions” should be fibrosis or fibrotic responses, depends on the content.

“Striking investigations” what do you mean?

“Robust analysis” what do you mean?

Line 207 “Increased expression for inflammatory cytokines and fibrotic matrices collagen underpinned the histolopathological investigations.” This is also an unclear sentence. For example: an increased expression cannot underpinned or expression of (not for).

 236

a low miR-29a signaling escalates COL3.1, increasing fibrotic matrix formation in

237

subacromial bursa in the progression of rotator cuff lesion with shoulder stiffness. Gain of miR-29a function

238

reverses tendon tissue injury attaining shoulder function. Profound analyses offer new molecular insight to

rotator cuff tear with shoulder stiffness

Author Response

Response to Reviewer #2

In the manuscript entitled: MicroRNA-29a Mitigates Subacromial Bursa Fibrosis in Rotator Cuff Lesion with Shoulder Stiffness Ko at al., sudy the functional involvement of miR-29a in tenocytes and fibrosis and relate their observation to decrease miR-29a in rotator cull tissue from patients with shoulder stiffness and controls. The study is overall interesting and relevant to the field. It is also corroborates to other studies showing the involvement of miR-29a in tissue fibrosis and other pathologies. Moreover, their study reinforce a therapeutic potential for miR-29a.

However, there are several issues/ critics that must be addressed before publication, which are listed here under.

Authors’ response: We are grateful for your constructive comments and suggestions, which prompt us to improve this article. We rewrote the text, figure legends and methodology, etc.

General comments:

Shoulder diseases are categorized between chronic and acute. The chronic is age associated and as such tissue histology differs from the acute condition. The author should specify on the condition they have studied. Moreover, shoulder diseases are different pathologies, the authors should be more specific to which pathology they studied. The tissue that was used in human is not clear, was it the shoulder impingement, which part of the tissue was used for molecular analysis. More details will help the reader. Since it is a highly heterogenic tissue, it can explain the huge variations between individuals that are found in the molecular analysis.

Authors’ response: We apologize for not writing the methodology in a more detail way so that make you not to get around. We rewrote the patient recruitment (Lines 282 – 285) and specimen harvest (Lines 295 – 298) in the revised version, now read as follows:

Patients with rotator cuff tears with (n = 22) or without (n = 35) shoulder stiffness who had symptoms of impingement for over 3 months along with magnetic resonance imaging (MRI) of complete rotator cuff tear, as well as required surgery for open acromioplasty, lysis of adhesions or rotate cuff repair were included in this study.

In the surgical lysis of adhesions or excision of adhesive subacromial synovium during rotator cuff repair, subacromial synovium specimens from greater tuberosity to coracoid within rotate cuff lesion were harvested for quantitative RT-PCR and immunohistochemistry. In a subset of experiment, 10 ml venous blood was drawn from each patient preoperatively to probe serum miR-29 expression levels.

The human shoulder anatomy and biomechanics are not to be found in a mouse or any animal walking on four legs. This should be changed throughout the paper. Therefore, the results in mouse show an effect of mir-29a in fibrotic tissue. Moreover, what is the tissue that was studied in mouse: muscle tendon. The histology differs from that in human.

Authors’ response: Thank you for bringing this to our awareness. We acknowledged the limitation of this study with regards to the difference between human rotator cuff tear and experimental rotator cuff injury in mice in the revised version (Lines 265 – 272), which now read as follows:

We acknowledge the limitation of this study that the anatomy or biomechanics of supraspinatus and infraspinatus tenotomy-induced rotator cuff injury in mice may not be fully extrapolated to human rotator cuff lesion with shoulder stiffness. Microsurgery-mediated supraspinatus injury in mice has been utilized to explain the cellular and molecular events underlying human shoulder disorders with rotator cuff tears [47,48]. Chronic inflammation and fibrosis are prominent features of this experimental rotator cuff injury model, which is consistent with the histopathology of fibrotic tissue overdevelopment in human shoulder stiffness. In this study, the experimental animal model further explains the biological function of miR-29a indispensable in protecting against fibrosis in the development of shoulder stiffness.

How quantification from histological images were made is not entirely clear. In principles, a colorimetric signal cannot be quantified. Trichome staining is not quantifiable. The authors write: Line 272: Number of cells positive for COL3a1 immunostaining and number of vascular tissue positive for a-SMA in 3 randomly selected fields in each section were counted.” The resolution in Fig. 2 and Fig 4 of those staining is insufficient for counting. Moreover, COL3a1 is secreted, who it is possible to count positive cells? As a note: Quantification of collagen fluorescence signal is much more reliable. In addition, fibrosis in the RC is associated with an increase in fat (detected by MRI) and an increase in fatty droplets (Nile Red staining), as signal that can also be quantified.

Author response: We rewrote the methodology for histomorphometry (Lines 300 – 311) in the revised version, which now read as follows:
Upon decalcifying in ethylenediaminetetraacetic acid-phosphate buffered saline mixture, subacromial bursa specimens were embedded in paraffin and sectioned for hematoxylin-eosin staining for gross histology. Fibrotic tissue in specimens was assayed using Masson’s trichrome staining (Sigma-Aldrich), according to the manufacturer’s manual. Thickness of subacromial bursa membrane and area of fibrotic tissue were microscopically analyzed using Zeiss microscope and image analysis software. Six fields of 2 sections of each human specimen and 36 fields of 12 sections from 6 murine specimens were randomly selected for histomorphometry. Fibrosis was expressed as the percentile of area of fibrosis/total area of interest [19]. For the detection of miR-29a transcript in specimens, in situ hybridization was performed using miR-29a probes (Applied Biosystems) labeled with digoxigenin along with digoxigenin antibody conjugated with horseradish peroxidase (Roche) [21]. In some experiments, collagen 3A1 (COL3A1) and α-smooth muscle actin (α-SMA) antibodies, as well as immunohistochemistry analysis kits containing IgG-conjugated with horseradish peroxidase (BioGenex) were utilized to detect fibrotic tissue and vessels in sections. Number of COL3A1 immunostained cells and number of vascular tissue positive for α-SMA in 3 randomly selected fields in each section were counted.

Specific comments:

Results and methods
Fig. 1 and 2 the variations are enormous, was a test for outliers was applied?

Author response: We used Wilcoxon test to analyze the difference between human specimens in stiffness and non-stiffness groups. Gender and number of involved shoulders in patients was analyzed by Chi-square test.

Fig. 1A, for the SMA staining, a different contrast and brightness was applied.

Author response: The image quality was improved in the revised version.

The statistical test should be specified in the legends not only in the results.

Author response: We revised the statistical analysis in each legend of figure in the revised version.

The presentation of the in situ hybridization results is not convincing. We also know that miRs are secreted and since they are part of cytoplasmic RNA decay signal other than nuclear was reported. Alternatively, images should be taken using a higher magnification.

Author response: Thank you for bringing this to our awareness. In situ hybridization images showed that cytoplasmic compartment in subacromial bursa showed miR-29a transcripts (brown). We improved the image resolution to make clear the miR-29a distribution.

The methods for the RT qPCR are not entirely clear and crucial details are missing in order to assess the results: how cDNA was made with oligo d(T) and/or random primer? Normalization to the 18S rRNA is often considered unreliable as the CT values are very low compared with mRNAs. The authors should show CT values. Moreover, it cannot be used for miRNA normalization. This requires another (unchanged) miRNA in combination with spike. At minimum the authors should show CT values and an unrelated miRNA with unaltered levels.

Author response: We re-wrote the methodology of reverse transcription (Lines 315 – 319) in the revised version, now read as follows:

1 g total RNA was pipetted for reverse transcription into cDNA using ReadyScript® Two-Step cDNA Synthesis Kits (Sigma-Aldrich), according to the manufacturer’s instructions. In brief, 1 g total RNA were mixed with 4 μl of ReadyScript cDNA Synthesis Mix with M-MLV reverse transcriptase, MgCl2, oligo-dT and random primers. The mixtures were incubated in 25℃ for 5 min, 42℃ for 30 min and 85℃ for 5 min.

We also re-wrote the RT protocols for assessing miR-29 (Lines 324 – 329) in the revised version, now read as follows:

For probing miR-29 expression, 1 mg total RNA was reversely transcribed using TaqMan® MicroRNA Reverse Transcription Kits with 10X RT buffer, dNTP mix, RNAase inhibitor and MultiScribe™ RT enzyme, according to the maker’s manuals. The RT mixtures were subjected to PCR amplification using 2× TaqMan® Universal PCR Master Mix and specific primers for miR-29a, miR-29b, miR-29c and housekeeping gene U6. Relative microRNA expression was calculated as 2-ΔCt, where ΔCt = CtmiR-29a – CtU6.

The in vivo near infrared fluorescence assay is not commonly used and could be elaborated in more details, also in the Results section. The authors refer to two different references, this is confusing.

Author response: We re-wrote the use of in vivo near infrared fluorescence approach to trace tissue inflammation (Lines 122 – 125) and methodology (Lines 349 - 354) in the revised version, now read as follows:

We adapted 2-deoxyglucose along with in vivo near infrared fluorescence imaging approachm, which has been utilized to trace inflammation in arthrosclerotic vessels, osteoarthritc synovium, and acute pancreatitis [24,25,26], to verify whether miR-29a overexpression change inflammation of injured rotator cuff.

Inflammation in lesion site in the involved shoulders was traced using near infrared fluorescence system, as previously described [24,25,26]. In brief, at 2 weeks postoperatively, mice were anesthetized and injected 10 nM 2-deoxyglucose (LI-COR Bioscience) via tail vein. The uptake of fluorescence 2-deoxyglucose was scanned using near-infrared fluorescence in vivo imaging system (Pearl Impulse IRDye 800CW, LI-COR Bioscience). Fluorescence intensity of region of interest (5 mm × 5 mm) of lesion site were quantified using a 774-nm absorption wavelength of the system, according to the manufacturer’s instruction.
Fig. 6 the effect on tissue histology is not very convincing: a treatment with miR0-29a induces fatty formation.

Author response:

The legend to Fig. 7 is mixed up or incomplete. The color-code of the bars is not specify. What are three repeated experiments? What is “forced expression” is it overexpression? Is this overexpression under physiological conditions? It is essential to show the expression of other genes (also house-keeping) in this overexpression condition. Mir-29a has over 1000 predicted target genes, in mirDB, assessing one gene is not sufficient to make a solid point. Exp. 7D should be explained in more details, the third bar (miR-29a +col3A1-3’UTR) is not to be seen, also not SD.

Author response: We rewrote the legend of Fig. 7 (Lines 202 – 212) in the revised version, which now read as follows:
Analysis of miR-29a action to human tenocyte cultures. IL-1β rather than IL-6 and IL-8 significantly Decreased miR-29a expression in tenocytes. Cell cultures were incubated in 1 ng/ml IL-1β, IL-6 or IL-8 for 24 hours (A). miR-29a expression was increased in cells upon miR-29a precursor transfection, whereas miR-29a antisense oligonucleotide (miR-29a-AS) transfection reduced the expression (B). miR-29 precursor transfection attenuated the IL-1β-mediated COL3A1 expression, whereas miR-29a knockdown alone provoked COL3A1 expression (C). Sequence of the miR-29a targeted 3’-untranalated region (3’-UTR) and 3-base pair mutant of COL3A1 were designed and constructed for luciferase reporter assay. miR-29a precursor transfection decreased luciferase reporter activity of 3’-UTR rather than mutant of COL3A1 in tenocytes, whereas miR-29a-AS transfer increased the luciferase reporter activity (D). Data are expressed as mean ± SEM calculated from 3 experiments analyzed using ANOVA test and Bonferroni post hoc test. Asterisks * indicate significant different between groups (P < 0.05).
We also re-wrote the rationale of the assessment of miR-29a targeting COL3A1 and acknowledged the limitation of this study (Lines 255 – 259) in the revised version, which now read as follows:

COL3A1 is an important hallmark of fibrosis in chronic rotator cuff tears [44,45]. In this study, miR-29a directly targeted COL31 3’-UTR, which underpinned its fibrosis-inhibitory action in subacromial bursa fibroblasts. We do not exclude the possibility that miR-29a may target other fibrogenic factors or fibrotic matrices in rotator cuff injury. The anti-fibrosis spectrum of miR-29a in rotator cuff injury warrants further investigations.

Discussion:

As the study shows the effect of miR-29a on fibrosis and Col1A expression, the discussion should focus on that. The relevance to shoulder disease is important, but indirect.

Author response: We re-wrote the rationale of the assessment of miR-29a targeting COL3A1 and acknowledged the limitation of this study (Lines 255 – 259) in the revised version, which now read as follows:

COL3A1 is an important hallmark of fibrosis in chronic rotator cuff tears [44,45]. In this study, miR-29a directly targeted COL31 3’-UTR, which underpinned its fibrosis-inhibitory action in subacromial bursa fibroblasts. We do not exclude the possibility that miR-29a may target other fibrogenic factors or fibrotic matrices in rotator cuff injury. The anti-fibrosis spectrum of miR-29a in rotator cuff injury warrants further investigations.

Nearly the whole first section of the Discussion is Background information and should be placed in the Introduction.

Author response: We rewrote the text in revised version. Discussion, methodology and figure legends are in particular.

Grammar:

The manuscript contains unclear terms and unclear sentences. Few are listed here under. But it should carefully checked.

Author response: We rewrote the text, corrected typos and invited a native speaker to revised the text.

Line 37: shoulder disorders (not one disorder).
Line 41 should be “are”.
Lines 202-203, not coherent.

Author response: We rewrote the text, corrected typos and invited a native speaker to revised the text.

The following sentence also does not make sense.
miRNA affect mRNA delay and hence mRNA levels. Gene expression is not a specific term.
“Fibrotic reactions” should be fibrosis or fibrotic responses, depends on the content.
Line 43
“Striking investigations” what do you mean? Line 223
“Robust analysis” what do you mean? Line 208
Line 207 “Increased expression for inflammatory cytokines and fibrotic matrices collagen underpinned the histolopathological investigations.” This is also an unclear sentence. For example: an increased expression cannot underpinned or expression of (not for).
Line 236 a low miR-29a signaling escalates COL3.1, increasing fibrotic matrix formation in
Line 237 subacromial bursa in the progression of rotator cuff lesion with shoulder stiffness. Gain of miR-29a function
Line 238 reverses tendon tissue injury attaining shoulder function. Profound analyses offer new molecular insight to rotator cuff tear with shoulder stiffness.

Author response: We rewrote the text, especially discussion, methodology and figure legends in the revised version. We changed “fibrotic reactions” to “fibrotic responses”. We also changed the wording, like striking investigations, robust analysis and profound analysis, etc. to relevant words.